# New Regenerative and Anti-Aging Medicine Approach Based on Single-Stranded Alpha-1 Collagen for Neo-Collagenesis Induction: Clinical and Instrumental Experience of a New Injective Polycomponent Formulation for Dermal Regeneration

**DOI:** 10.3390/biomedicines12040916

**Published:** 2024-04-20

**Authors:** Luigi Di Rosa, Antonino De Pasquale, Sara Baldassano, Noemi Marguglio, Patrik Drid, Patrizia Proia, Sonya Vasto

**Affiliations:** 1Department of Biological, Chemical and Pharmaceutical Sciences and Technologies (STEBICEF), University of Palermo, 90100 Palermo, Italy; sara.baldassano@unipa.it (S.B.); nmarguglio97@gmail.com (N.M.); sonya.vasto@unipa.it (S.V.); 2Independent Researcher, 95100 Catania, Italy; info@dottordepasquale.it; 3Faculty of Sport and Physical Education, University of Novi Sad, 21000 Novi Sad, Serbia; patrikdrid@uns.ac.rs; 4Department of Psychological, Pedagogical and Educational Sciences, Sport and Exercise Sciences Research Unit, University of Palermo, 90100 Palermo, Italy; patrizia.proia@unipa.it; 5Euro-Mediterranean Institutes of Science and Technology (IEMEST), 90139 Palermo, Italy

**Keywords:** regenerative medicine, KARISMA, skin regeneration, hyaluronic acid, human recombinant collagen, carboxymethyl cellulose, skin rejuvenation, Antera 3D skin scanner, neocollagenesis

## Abstract

This study explores the efficacy of a novel polycomponent formulation (KARISMA Rh Collagen^®^ FACE, Taumedika Srl, Rome, Italy), containing 200 mg/mL of non-crosslinked high-molecular-weight hyaluronic acid (HMW-HA), 200 μg/mL of a human recombinant polypeptide of collagen-1 alpha chain, and 40 mg/mL of carboxymethyl cellulose (CMC) as a regenerative medicine for skin regeneration and rejuvenation. This formulation combines non-crosslinked high-molecular-weight hyaluronic acid, human recombinant polypeptide of collagen-1 alpha chain, and carboxymethyl cellulose to stimulate collagen type I production and enhance skin hydration. This study involved 100 subjects with varying skin conditions, divided into three groups based on skin aging, smoking history, and facial scarring, to evaluate the product’s effectiveness in skin regeneration and aesthetic improvement. The methodology included two injections of Karisma (2 mL for each injection) one month apart, with evaluations conducted using FACE-Q questionnaires, the SGAIS Questionnaire, and Antera 3D skin scanner measurements at baseline, 30 days, and 60 days post-treatment. The results demonstrated a significant reduction in skin roughness and an improvement in skin quality across all the groups, with no correlation between the outcomes and the patient’s age. The subjective assessments also indicated high satisfaction with the treatment’s aesthetic results. The analyzed data allow us to conclude that the single-stranded collagen with hyaluronic acid and carboxymethyl-cellulose formulation is able to stimulate the skin’s regenerative response, yielding significant results both in vitro and, through our study, also in vivo. This new polycomponent formulation effectively stimulates skin regeneration, improving skin quality and texture, with significant aesthetic benefits perceived by patients, and a low incidence of adverse events, marking a promising advancement in regenerative medicine.

## 1. Introduction

Over the past decade, the field of regenerative medicine has made significant progress, particularly in skin treatment and rejuvenation. One of the major problems linked to the use of collagen has always been that of its immunogenicity, for which the safety profile of collagen has always been more risky than that of hyaluronic acid [1,2]. In the past, in fact, collagen derived from bovines and pigs possessed a series of limitations linked to the onset of allergic phenomena or possible infections such as spongiform encephalopathies [1,2,3,4]. The aim of this work was to evaluate for the first time, in vivo in humans, the regenerative and anti-aging capacity of a new injective formulation based on recombinant human collagen that allows us to avoid all the problems that in the past were connected with the use of collagen of animal origin.

During the aging process, both the epidermis and dermis undergo degenerative changes and the dermis shows the most clear changes. In general, with advancing age, the accumulation of UV (ultraviolet rays) damage and environmental pollutants leads to skin changes, with a progressive thinning of the epidermis with wrinkle formation, loss of elasticity, and hyperpigmentation phenomena, collectively known by the terms “chrono-aging” and “photo-aging”. Wrinkles and reduced elasticity are the result of the progressive atrophy of the dermis, linked to changes in the extracellular matrix (ECM), and in particular, the collagen quantity in the dermis. Unlike the epidermis, which is made up of keratinocytes, the dermis is mainly composed of an acellular component; collagen fibers are the main component of the ECM, accounting for 75% of the skin’s dry weight, and provide tensile strength and elasticity. In human skin, type I collagen makes up 80–90% of the total collagen, while type III makes up 8–12% [5].

This article aims to explore in detail the effectiveness of a new injective polycomponent formulation (KARISMA Rh Collagen^®^ FACE, Taumedika Srl, Rome, Italy) in dermal regeneration.

KARISMA, hereinafter described only with the letter “K”, contains 200 mg/mL of non-crosslinked high-molecular-weight hyaluronic acid (HMW-HA), 200 μg/mL of a human recombinant polypeptide of collagen-1 alpha chain, and 40 mg/mL of carboxymethyl cellulose (CMC). The combination of these molecules aims to stimulate epidermal regeneration by collagen type I production, as well as skin hydration [5]. In vitro studies have shown that the human recombinant polypeptide of collagen-1 alpha chain, present in K, has an important effect in collagen 1 deposition both in young and in aged fibroblast cell cultures. Furthermore, TGF-β1 is increased in cell culture after K exposure [6]. TGF-β1 is a significant regulator of extra cellular matrix (ECM) activities, controlling the production of matrix metalloproteinases (MMPs) and serving as the primary regulator of collagen synthesis [7].

K’s human recombinant collagen α1 chain is produced with a patented technology that uses transgenic silkworms that promote the synthesis of the alpha helix filament within their cocoons. It is not immunogenic, and has a similarity to human collagen of 99.9%. Compared to its predecessors of animal origin, it does not present a clinically relevant allergic potential. Although collagen had already been used successfully in the past [1,2,3,4,8,9,10,11,12], it was burdened by a certain rate of adverse events such as allergic reactions and granulomas [4,5,13,14,15] and, therefore, it was almost completely abandoned as an injective device. However, recent studies have highlighted how new collagen production techniques make it suitable and safe as an injectable substance and can stimulate the skin’s natural regeneration processes, inducing the endogenous production of new collagen and elastin fibers, key elements for the recovery of elasticity and skin structure [2].

In addition, hyaluronic acid, present in K, is a widely studied and used molecule present in the fields of dermatology, aesthetic medicine, cosmetology, but also orthopedics and ophthalmology [14]. Hyaluronic acid is known for its hydrating properties and for its ability to stimulate the production of collagen, improving the elasticity and texture of the skin. Over the last twenty years, the scientific literature has widely documented the effectiveness of injectable hyaluronic acid in the treatment of wrinkles and in improving the general appearance of the skin, highlighting excellent results both in terms of safety and effectiveness. Its ability to retain large amounts of water also contributes to a “plumping” effect that visibly reduces wrinkles and improves skin texture [14,15].

Finally, the introduction of hydroxymethylcellulose in this formulation represents a significant innovation. Carboxymethylcellulose (CMC) is an FDA-approved water-soluble polysaccharide, derived from cellulose [14]. This component not only acts as a stabilizer, but also enhances the effectiveness of the other two ingredients, ensuring a more homogeneous distribution and the longer-lasting impact of the treatment, among other things. Delaying the action of hyaluronidase translates into a more marked and long-lasting regenerative effect, with both subjective (patient perception) and objective (clinically measurable) benefits.

In this study, the effects of this injective polycomponent formulation were examined in a cohort of 100 subjects with different skin conditions. The cohort was divided into three groups: a group of 50 patients included people affected by skin aging of various degrees, a second group of 20 subjects included patients who had smoked for at least 10 years, affected by skin aging, and a final group of 30 patients were suffering from facial scarring. The objective was to evaluate not only the effectiveness of the product in terms of the aesthetic result perceived by the patient, but also its role in skin regeneration, offering a significant contribution to the field of regenerative medicine. To verify these hypotheses, we used a validated facial skin scanner (ANTERA 3D Skin Scannera—Miravex Limited, Dublin, Ireland) [16,17] to evaluate how the depth of skin wrinkles changed, to evaluate the concentration of melanin, and to evaluate the concentration of hemoglobin, and, at the same time, we assessed through psychometric tests (FACE-Q Questionnaires and SGAIS Questionnaires) how patients assessed the improvement in skin parameters.

## 2. Materials and Methods

### 2.1. Inclusion Criteria

All patients had to be in good general health, without cardiovascular or metabolic disorders; patients did not have to take medications, and oral supplements including vitamins, hyaluronic acid, collagen, zinc, keratin were not permitted during the study. Patients could use makeup remover wipes to remove makeup but had to refrain from using moisturizing creams or restorative cosmetic creams during the study period. Patients must not have received any injection treatments (biorevitalizers, biostimulants, hyaluronic acid, calcium hydroxyapatite, botulinum toxin) on the face or neck for at least six months before the study to be eligible for enrollment.

### 2.2. Exclusion Criteria

Patients with active dermatological diseases such as herpes, folliculitis, seborrheic dermatitis, eczema, psoriasis; patients with rheumatic or immunological diseases, cancer patients, pregnant or breastfeeding patients; and patients with allergic diathesis to drugs or foods were all excluded.

### 2.3. Patient Selection and Study Design

For the present study, the consent of the ethics committee was requested and obtained (Ethics Committee—Commission University of Novi Sad-Serbia, Decision No. 49-10 February 2023; Approved). The present study was also included and approved as an American government clinical trial (ClinicalTrials.gov ID: NCT06152718).

A total of 145 patients were initially enrolled on the present study; however, for some of them, it was not possible to perform all the subjective assessments (Face-Q questionnaire and Subject Global Aesthetic Improvement Scale (SGAIS)) and objective assessments through the 3D scanner and were therefore removed from the final version of the statistical analysis. Overall, 100 patients completed this study and were analyzed. The age of the patients ranged from 27 to 70 years and were of both sexes, divided into three subgroups of patients. One group of non-smoking patients was composed of 50 (age range: 30–70 years; mean 49.76 ± 10.916; 42 females—8 males) subjects affected by skin aging of various degrees (normal group); one group of smoking patients was composed of 20 (age range: 31–70 years; mean 52.5 ± 13.233; 18 females—2 males) subjects who had smoked for at least 10 years and suffered from various degrees of skin aging (smoker group); one group of 30 (age range: 27–48 years; mean 40.17 ± 4.92; 27 females—3 males) subjects suffered from acne scars on their faces (acne group). Descriptive statistics of the groups under analysis are summarized in Table 1. All procedures involved in the study were explained in detail and written informed consent was obtained from all subjects.

### 2.4. Evaluation

Each subject underwent the same protocol, based on two injections of K one month apart from each other as per the manufacturer’s technical data sheet.

This injective polycomponent formulation is a new proprietary formulation that includes high-molecular-weight hyaluronic acid (HA), human recombinant collagen α1 chain, and carboxymethylcellulose (CMC) in a 2 mL pre-filled syringe. Each subject was evaluated before starting the study and clinical efficacy assessments were conducted at 30 and 60 days after baseline. For each patient, adverse events were also recorded. In particular, during the preliminary visit, the patient was subjected to a series of multiple-response tests already validated by international scientific literature (FACE-Q questionnaires) to evaluate the patient’s subjective consideration of their condition before treatment. In the same session, a skin evaluation was performed using the Antera 3D skin scanner tool (Miravex Limited, Dublin, Ireland) to assess the presence of fine, medium, and large wrinkles on the skin surface and were expressed in millimeters; Antera is able to detect and measure skin roughness from 1 mm to 4 mm and above, so investigators were able to analyze and measure all skin wrinkles on the skin surface. Using the Anter 3D scanner, melanin and hemoglobin quantity were also evaluated, with values expressed in absolute numbers.

During the first session (T0), each patient completed Face-Q questionnaires, underwent measurements using the Antera scanner, and received the first injection in the face, neck, and periocular region. For each patient, only one syringe was used for each session, corresponding to 2 mL of product, which was distributed uniformly in the areas under examination.

The same protocol was repeated one month later (T1), when the patients were again subjected to the same type of questionnaire (FACE-Q questionnaires), and another scan was performed with the Antera 3D skin scanner. Furthermore, to evaluate the overall degree of global aesthetic improvement after first injection, the Subject’s Global Aesthetic Improvement Scale (SGAIS) questionnaire was also administered to the patients, and they were then subjected to the second injection of the injective polycomponent formulation.

A duration of 30 days after the second injection (T2), the patients received the last check-up, the FACE-Q and SGAIS questionnaires were repeated and scans were performed again using the Antera 3D skin scanner.

The protocol can be summarized as follows:

T0: pre-assessment with Face-Q Questionnaire; first evaluation with Antera 3D skin scanner using 1 mm, 2 mm, and 4 mm filters in order to deeply evaluate all skin wrinkles; melanin quantities and hemoglobin quantities were also recorded; first 2 mL injection into the dermis approximately 1 mm deep in the face and neck.

T1: a total of 30 days after the first injection of the injective polycomponent formulation; second evaluation with Face–Q Questionnaire, second evaluation with Antera 3D skin scanner repeating all analyses already performed in T0; first evaluation with the SGAIS Questionnaire; second 2 mL injection into the dermis of the face and neck, in the areas treated during the first session.

T2: a total of 30 days after the second injection of the injective polycomponent formulation (60 days after the first injection); third evaluation with Face–Q Questionnaire, third evaluation with Antera 3D skin scanner, repeating all analyses performed in the previous two sessions; second evaluation with the SGAIS Questionnaire.

The FACE-Q [18,19,20,21] is a validated questionnaire that includes a set of more than 40 independently functioning scales and checklists that evaluate numerous aspects of the patient’s face, skin quality, and appearance. Depending on the procedure used, only the FACE-Q scales relevant to a particular patient or procedure need to be used. For the study in question, the following scales were used:

FACE-Q Age Appearance Appraisal; FACE-Q Face Overall; FACE-Q Lines Overall; FACE-Q Neck; FACE-Q Skin. Each questionnaire has a score ranging from 1 to 4 for each question. The overall score is obtained by adding the scores of the individual questions.

The FACE-Q Age Appearance Appraisal is a questionnaire composed of seven questions that enquire how a person feels about the age their face looks (overall score 7 to 28).

FACE-Q Face Overall is a questionnaire composed of 10 questions that enquire how a person feels about their satisfaction or dissatisfaction with their entire face (overall score 10 to 40). FACE-Q Lines Overall is a questionnaire composed of 10 questions that enquire how a person feels about the wrinkles on the entire face in both static and dynamic conditions (overall score 10 to 40). FACE-Q Neck is a questionnaire composed of 10 questions that aim to know how a person feels about the appearance of their neck in both static and dynamic conditions (overall score 10 to 40). FACE-Q Skin is a questionnaire made up of 12 questions that aim to find out how a person feels about the appearance of their skin’s complexion in terms of hydration, texture, tone, but also attractiveness towards others (overall score 12 to 48).

The FACE-Q|Aesthetics© is intellectual property of Drs Anne Klassen, Andrea Pusic, and Stefan Cano. The FACE-Q|Aesthetics© is owned by Memorial Sloan Kettering Cancer Center (New York City, NY, USA).

The Subject Global Aesthetic Improvement Scale (SGAIS) [22,23,24,25,26,27], is a validated scale used to evaluate the aesthetic improvement perceived by patients after cosmetic or surgical treatments. The SGAIS allows patients to rate their cosmetic improvement on different levels, usually from “worse” to “much improved”. This type of scale can be considered a variant of the Likert scale; however, the SGAIS is specifically designed for assessments in the field of aesthetics and may have a customized rating scale and terminology to better fit this context.

It is made up of five points, from one to five, where one represents worsening compared to the baseline, while five is the maximum possible improvement and allows the patient to verify the overall improvement by comparing the patient’s appearance after each injection session compared to baseline, as shown in Table 2.

The Antera 3D scanner [28,29,30,31,32,33,34] (Miravex Limited, Dublin, Ireland) is a handheld camera with a measuring area of 5.6 × 5.6 cm^2^. The instrument relies on technology related to photometric stereo measurements to reconstruct the skin in 3D using multiple images taken under different light sources with 7 different wavelengths spanning most of the visible spectrum. It allows, in a single acquisition, to analyze multiple parameters of skin shape and pigmentation. The Antera 3D v1.0 software allows you to analyze and measure the topographic characteristics of the skin, such as wrinkles and the mapping of hemoglobin and melanin and their relative concentrations. Since the camera opening is placed directly onto the skin, images are not affected by surrounding lighting conditions. Through the Antera 3D scanner, it is possible to evaluate the depth of wrinkles with great accuracy and it is possible to modify the analysis based on the depth of the wrinkles you want to analyze. In this study, three measurement filters have been set: (1) Small wrinkles (1 mm filter), used for fine lines. (2) Medium wrinkles (2 mm filter), used for average lines. (3) Large wrinkles (4 mm filter), used for deep wrinkles. The average melanin value was also analyzed, which indicates the average concentration of melanin in the selected area (arbitrary unit from 0.1–1), and the average hemoglobin value, i.e., the average concentration of hemoglobin in the selected area (arbitrary values ranging from 0.1 to 4).

For the study, three areas were chosen for each patient to be analyzed: an area at the level of the cheek called “cheek”, one at the level of the periocular region called “eye”, and an area at the level of the neck called “neck”. For the group of patients suffering from acne scars (acne group), the measurements were always performed on the cheeks “cheek” in an anatomical point where there were evident acne scars in order to follow any improvement in the scars over time.

In order to standardize the measurements, a circular selection with a 2.2 cm diameter circle was always used when using the Antera 3D v1.0 software supplied. The Antera 3D v1.0 software automatically calculates the parameter values in the selected circle area.

The Antera 3D scanner is able to standardize the areas examined. In fact, it uses a proprietary algorithm that automatically records two or more images by parameterizing them together. This automatic matching procedure allows for the automatic compensation of the rotational shifts of the images with the certainty that investigators are analyzing exactly the same portion of skin with the same dimensions over time.

### 2.5. Injection Technique

All patients underwent the same injection technique, which included multiple injections into the dermis approximately 1 mm deep at a regular distance of approximately 1 cm for each injection site with a micro-wheal technique over the entire face and neck. Each session, one syringe of the injective polycomponent formulation was used, consisting of a pre-filled sterile 2 mL syringe using the 27 gauge needles provided in the package. A 27 gauge needle was used for one side and the other needle provided in the package was used for the contralateral side in order to minimize tissue trauma; 0.1 mL of product was injected, on average, for each injection site and the patient was warned that the wheal on the dermis would remain for approximately forty-eight hours.

### 2.6. Statistical Analysis

All statistical analyses were performed using the International Business Machines (IBM) Corporation’s SPSS Statistics, version 25.0 (New York, NY, USA). All patients completed the questionnaires and instrumental evaluation during the first visit before treatment (T0), and then repeated the same questionnaires and instrumental evaluation at the first follow-up visit before treatment with second injection (T1), and finally, 30 days after the second injection (T2). Descriptive statistics for continuous variables were measured as minimum, maximum, mean, and standard deviation; discrete or categorical variable was measured as frequency; Table 1 summarizes the descriptive statistical values for the three analysis groups. Considering the sample size, non-parametric tests were used. The Friedman test for repeated measures was used to test the statistical significance of continuous variables such as Antera 3D scanner measurements; Kendall’s tau statistical index was used to test the correlations between variables and Wilcoxon Signed-Rank Test was used to evaluate pairwise comparisons.

## 3. Results

Using the Antera 3D scanner, the skin roughness values (wrinkle depth) were evaluated using three different cut offs: 1 mm filter (fine wrinkles), 2 mm filter (medium wrinkles) and 4 mm filter (big wrinkles), at T0, T1 and T2. The greater the roughness of the skin for each type of filter used, the greater the value detected by the scanner. Each filter examines only the wrinkles that reach up to and not beyond the corresponding value; therefore, the 1 mm filter (fine wrinkles) examines only the wrinkles with a maximum depth of 1 mm, excluding the others, and so on. In order to evaluate whether K had the ability to produce the significant regeneration of skin tissue, the differences between the skin roughness values were evaluated using the Friedman test, a non-parametric statistical test used to analyze the differences between more than two related groups.

The data analysis showed an evident progressive reduction in the mean values between the baseline (T0) and last follow-up at 60 days (T2), as can be seen in the boxplot examples shown in Figure 1, Figure 2 and Figure 3, in which the box plots using the 1 mm filter are shown. It is clear how the values tend to decrease from the first measurement (T0) to the last follow-up (T2). The values for the other filters used show the same clear trend (box plots not shown); therefore, to evaluate the statistical significance, the Friedman test for repeated measures was used, and it showed a high significance (*p* < 0.000) for all the groups analyzed, and for all the filters for skin wrinkles utilized, with the exception of the values found in the evaluation of the neck in the group of non-smoking patients (NORMAL GROUP) using the 2 mm filter (*p* = 0.771); data are summarized in Table 3.

The same evaluation method was also used to analyze the data related to the melanin and hemoglobin concentrations found. In this case, the data showed a trend that is not equally clear and defined, and, in fact, the Friedman tests showed that although the hemoglobin values decrease in the normal group from time T0 to time T2, there is no statistical significance (*p*: 0.749 for measurements of the cheeks; *p*: 0.015 for measurements of the eye, *p*: 0.111 for measurements of the neck), while the values for the group of smokers and the acne group showed a positive statistical significance (*p* < 0.000). The melanin concentration values showed a high significance too (*p* < 0.000), with the exception of the NECK values in the normal group (*p*: 0.207). The data are summarized in Table 4.

Furthermore, it was hypothesized that age could be a factor to consider in evaluating the effectiveness of K. For this reason, the possible correlation between the age of the patients and the results of the measurements for fine wrinkles (1 mm filter), for medium wrinkles (1 mm filter), and for large wrinkles (4 mm filter) through Kendall’s tau correlation analysis was examined. Kendall’s tau is the most used non-parametric correlation coefficient that measures the association between two variables. The correlation analysis demonstrated that the results of the measurements were not correlated with age; therefore, in none of the groups analyzed did the age factor appear to be able to modify the results of Karisma. The same type of analysis was performed between age and the concentration of melanin and hemoglobin, and even in this case, there were no statistically significant correlations. The results of these statistical analyses are summarized in Table 5.

In addition to the analysis of the objective data measured with the scanner, the subjective data perceived by the patients with the FACE-Q and SGAIS questionnaires were also evaluated. For the responses of the FACE-Q questionnaires, the Frieman test for repeated measures was used, while the Wilcoxon test for paired data was used for the responses of the SGAIS. The subjective questionnaires, both the Face-Q and the SGAIS, showed a high significance for each type of question proposed. The data relating to the Face-Q questionnaire are reported in Table 6 for the FACE-Q questionnaire and Table 7 for the SGAIS questionnaire. During the follow-up, the complications were also registered. A total of 35 patients (35%) had at least one or more than one hematoma due to sharp needle injection. A total of 15 patients had moderate swelling that lasted more than two days, but all resolved spontaneously within seven days. No major complications occurred. In particular, we did not find any intravenous, arterial, or venous injections and no extrinsic compressions, so we did not find any cases of skin necrosis or embolisms.

## 4. Discussion

Since the beginning of the 1970s, research has focused on the medical and surgical uses of collagen in various surgical specialties. Knapp and colleagues [4] studied the behavior of solubilized collagen as a bioimplant and Stegman and Tromovitch [9] focused on its first use for acne scars on 35 patients. Then, Kaplan [6], Cooperman [5], and Nicolle [8] studied collagen to correct a wide variety of clinical conditions, from post-surgical atrophy to acne or post-smallpox scars. Zeide [35], in 1986, and then, Clark [36], Matti [12], Stegman [13], and Lee [37], thoroughly analyzed the possible adverse reactions to collagen, calculating an overall incidence of adverse reactions to be between 3% and 5%, concluding that the use of bovine collagen was an absolute contraindication in patients with autoimmune diseases, atopy, or a clinical history of anaphylactic reactions. Charriere and colleagues [38] reported a strong correlation between the presence of antibodies against collagen and a positive response to the skin test (92%) or an adverse reaction (100%). Rapaport and colleagues [11] stressed the fact that a first test dose should be placed in the forearm and the response should be measured no earlier than 4 weeks after, so before actually performing the injections, patients had to wait at least 30 days. Moon [39] tested the efficacy of porcine collagen versus bovine collagen, concluding that porcine collagen had a similar efficacy but was free from the risk of bovine spongiform encephalopathy.

Burke and colleagues [10] were the first to prove that collagen could have regenerative effects on the dermis; the results of this study suggested that the bovine implant material stimulated a host response resulting in the degradation of the injected filler and its replacement with newly generated human collagen. However, in 1998, Olenius [40] published a clinical study using a new biodegradable implant for the treatment of lips, wrinkles, and folds, evaluated according to the new EN 540 directive for medical implants. The product was the first filler based on hyaluronic acid stabilized by crosslinking. From that moment on, although collagen was also studied in the following years [41,42,43], hyaluronic acid began its unstoppable rise [44,45,46].

The regenerative capabilities of K had already been observed, in vitro, by Augello and colleagues [2], using cell lines of normal human fibroblasts placed in culture (Normal Human Dermal Fibroblasts—NHDFs) and cell lines of aged fibroblasts (Aged Human Dermal Fibroblasts—AHDFs). The results demonstrated that fibroblasts responded to K by increasing cell growth in a time- and concentration-dependent manner; the higher the concentration of k, the greater the cell proliferation, with a concentration of 5% proving to be the most effective at stimulating the proliferation of both normal and aged fibroblasts. In healthy fibroblasts, K increased cell growth; in fibroblasts in which the aging process was induced, it was able to counteract hydrogen peroxide-induced senescence in a concentration-dependent manner, and even the expression of intracellular collagen type one was significantly up-regulated in fibroblasts exposed to K, as were the levels of transforming growth factor beta (TGF-β1). Furthermore, in a simple wound healing model, it was noted that the cellular migration of fibroblasts, defined as the time taken to close an open lesion after a linear “scratch” that is artificially created on a monolayer of cultured cells, showed the closure speed at 6, 12, and 24 h increased in normal fibroblasts treated with K compared to the control in a concentration-dependent manner.

To the best of our knowledge, however, there has been no study analyzing the regenerative properties of K in vivo on human patients. The aim of our work was to evaluate the capabilities of K in vivo on human patients, in order to verify whether the regenerative capabilities demonstrated on fibroblasts could also be translated into measurable skin changes using accurate, repeatable, and validated instruments such as the Antera scanner. Furthermore, it was equally important that any changes in the quality of the skin were also perceived by the patient, thus making the positive changes induced by the regeneration of the skin tissue evident. The data showed a clear decrease in skin roughness for each group analyzed, with a high statistical significance for both mild, moderate, and large wrinkles. The same significance was also found in the remodeling of acne scars, which showed a clear decrease in the size and depth of the scars, following the tissue regeneration stimulated by K. The data obtained are significant in every anatomical area taken under examination, both in the face, periocular region, and neck; the only nonsignificant results were found in the neck measurements for average wrinkles in the group of 50 patients, and for the hemoglobin values, for which the trend was not always clear and therefore was considered by the authors to be overall not statistically significant. Examples of the effect of K from baseline up to time T2 can be seen in Figure 4, Figure 5, Figure 6 and Figure 7.

Furthermore, the data show that K elicits a good response regardless of the patient’s age, stimulating a regenerative effect on the skin. Another particularly interesting finding concerns the melanin values, which seem to decrease over time. Although at first, it may seem counter-intuitive that a stimulator of collagen proliferation could have some biological effect on the production of melanin, several recent scientific studies have shown a clear correlation between collagen and melanin. In fact, many authors have demonstrated how fibroblasts and, precisely, TGF-β1, together with other chemical mediators, have a role in melanic induction and metabolism [47,48,49], and Augello and colleagues [2] demonstrated in vivo that K has a direct and concentration-dependent effect on the increase in TGF-β1. The improvements produced by K are clearly perceived by the patients who responded, demonstrating high statistical significance for the questions of both the Face-Q and SGAIS validated tests.

The present study has intrinsic limitations, according to the authors, linked to the number of enrolled patients, which, although is a fairly adequate number, should be expanded in further studies confirming the findings of the authors; moreover, the follow-up appears to be quite short although the findings were very significant, so further studies should evaluate the long-term effects of this product.

## 5. Conclusions

The formulation of K has led to significant results both in vitro and, through our study, also in vivo. The data show that the single-stranded collagen with hyaluronic acid and carboxymethylcellulose formulation is able to stimulate the skin’s regenerative response. The changes in skin quality are objective and measurable with ease and repeatability, and the results also imply that it is perceived by patients as an effective treatment, leading to aesthetic results which are greatly appreciated by patients, without, at least in our series, adverse events, which have been described in numerous previous studies. The presence of human recombinant collagen has been shown to have no relevant immunogenic characteristics and to have a safety profile equal to that of hyaluronic acid, as demonstrated by the lack of adverse events in our population sample, allowing us to start using collagen again as a valid tool in regenerative and anti-aging medicine, although subsequent studies are necessary to evaluate the effectiveness of the results in the long term.

## Figures and Tables

**Figure 1 biomedicines-12-00916-f001:**
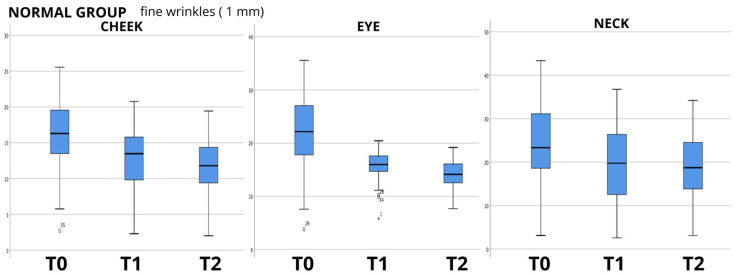
Box plot graphs for “normal group” (non-smoker patients) showing the values of skin wrinkle depth using 1 mm filter in the three anatomical areas examined (cheek—eye—neck) and how they change over time from baseline (T0) to last follow-up (T2).

**Figure 2 biomedicines-12-00916-f002:**
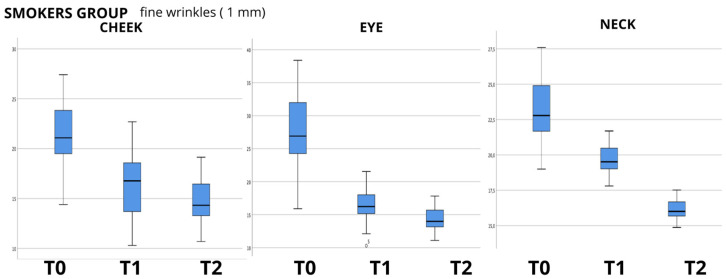
Box plot graphs for “smoker group” (smoker patients) showing the values of skin wrinkle depth using 1 mm filter in the three anatomical areas examined (cheek—eye—neck) and how they change over time from baseline (T0) to last follow-up (T2).

**Figure 3 biomedicines-12-00916-f003:**
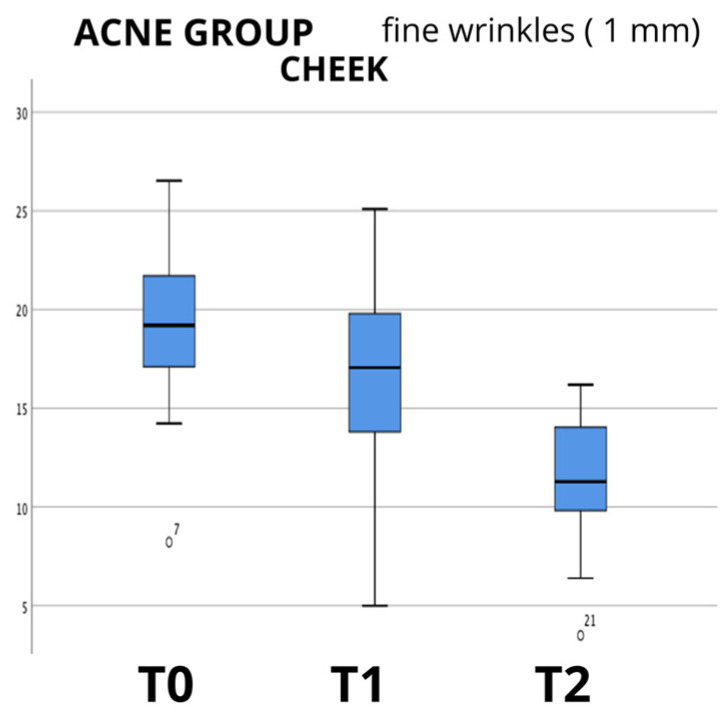
Box plot graphs for “acne group” (patients with acne scars) showing the values of skin examined in cheek area only, where scars were much more evident, using 1 mm filter, and how they changed over time from baseline (T0) to last follow-up (T2).

**Figure 4 biomedicines-12-00916-f004:**
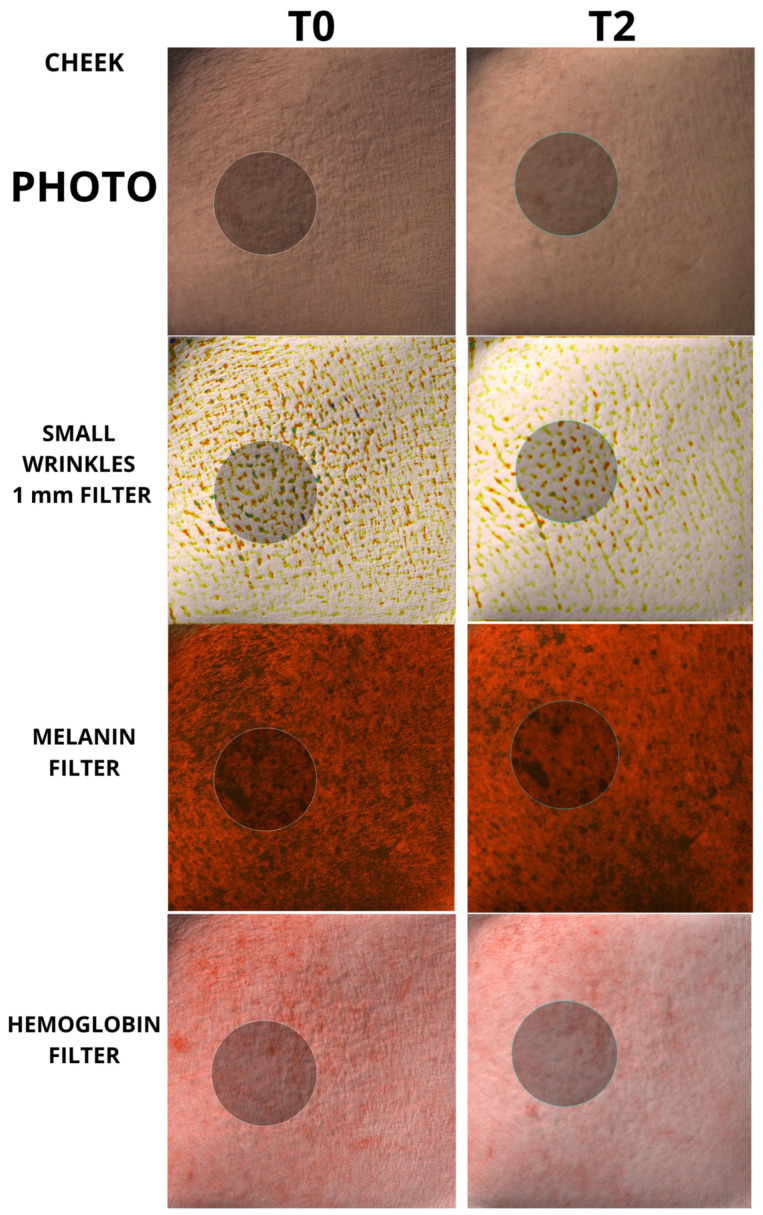
Example of automatic Antera 3D matching algorithm for finding the exact position during different measurements. Picture taken from cheek area (cheek). A 2.2 cm circle has been used for each measurement. Baseline and T2 pictures are shown.

**Figure 5 biomedicines-12-00916-f005:**
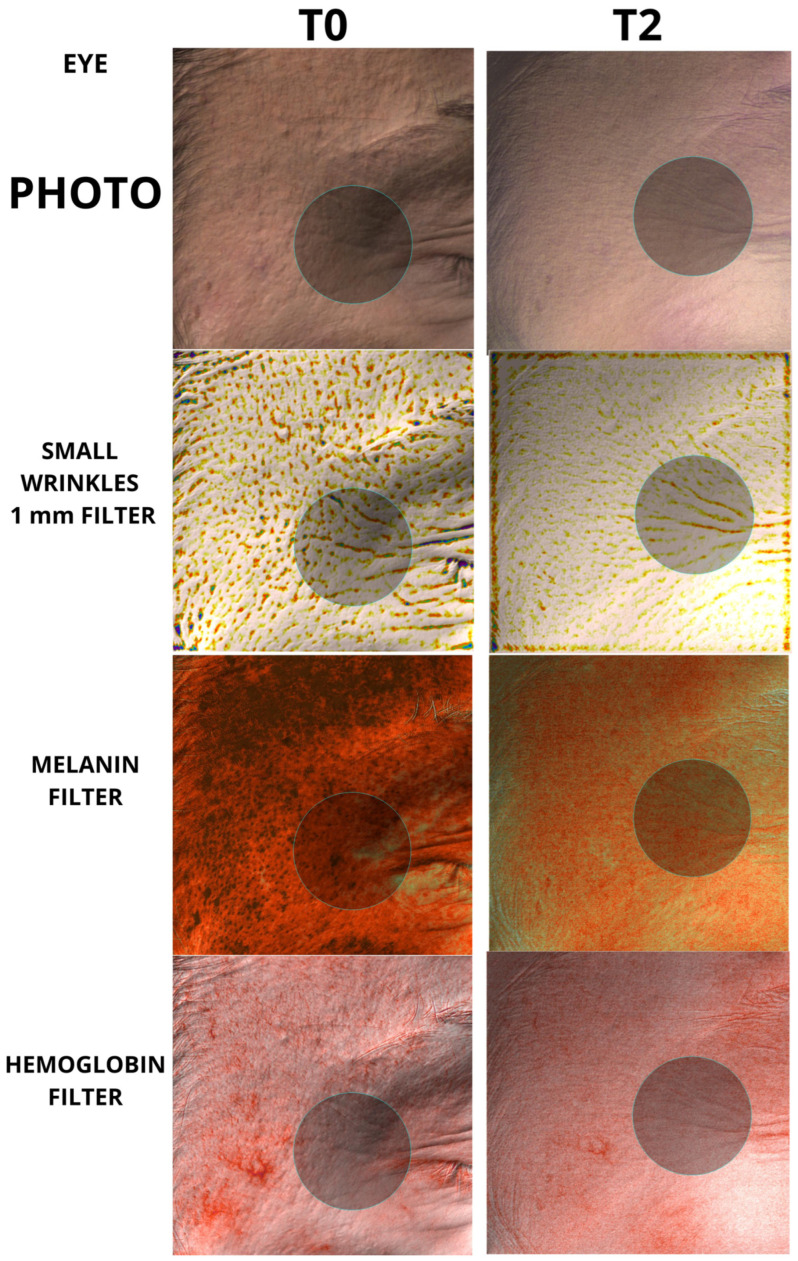
Picture taken from periocular area (eye). A 2.2 cm circle has been used for each measurement. Baseline and T2 pictures are shown.

**Figure 6 biomedicines-12-00916-f006:**
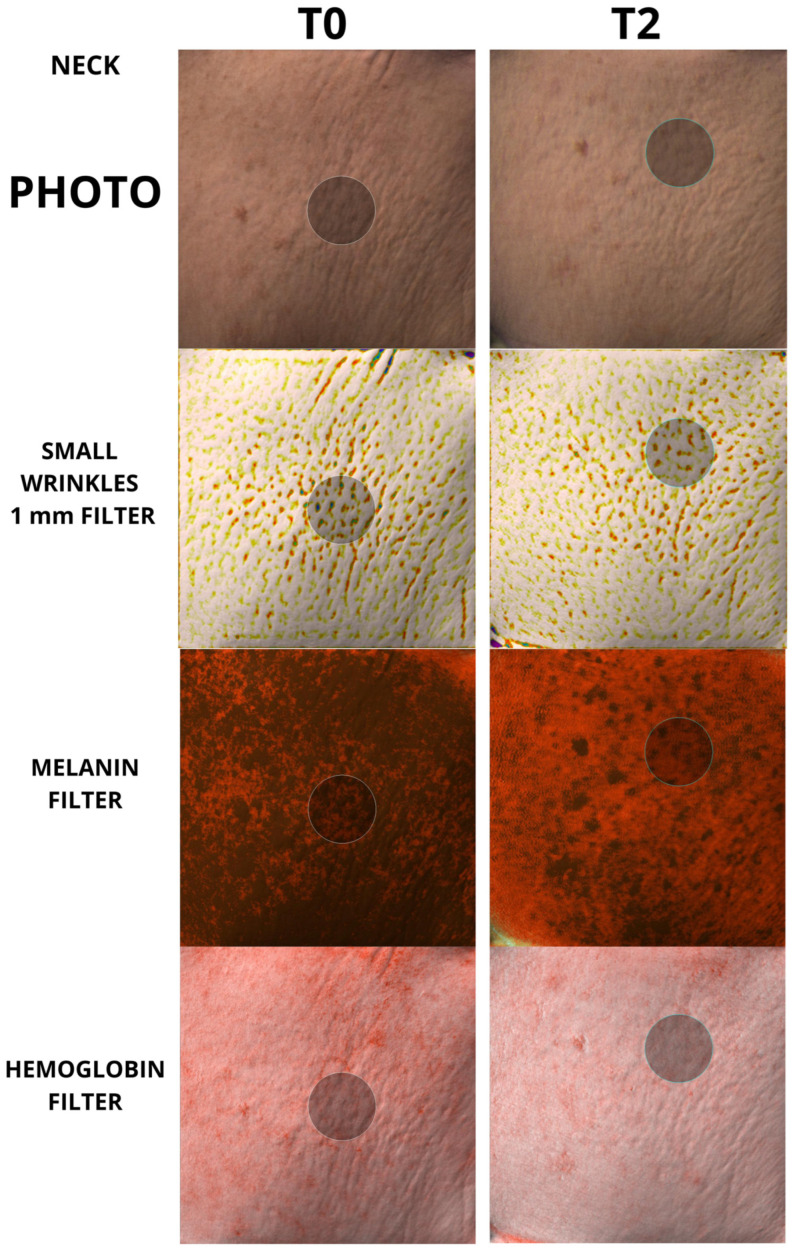
Picture taken from neck area (neck). A 2.2 cm circle has been used for each measurement. Baseline and T2 pictures are shown.

**Figure 7 biomedicines-12-00916-f007:**
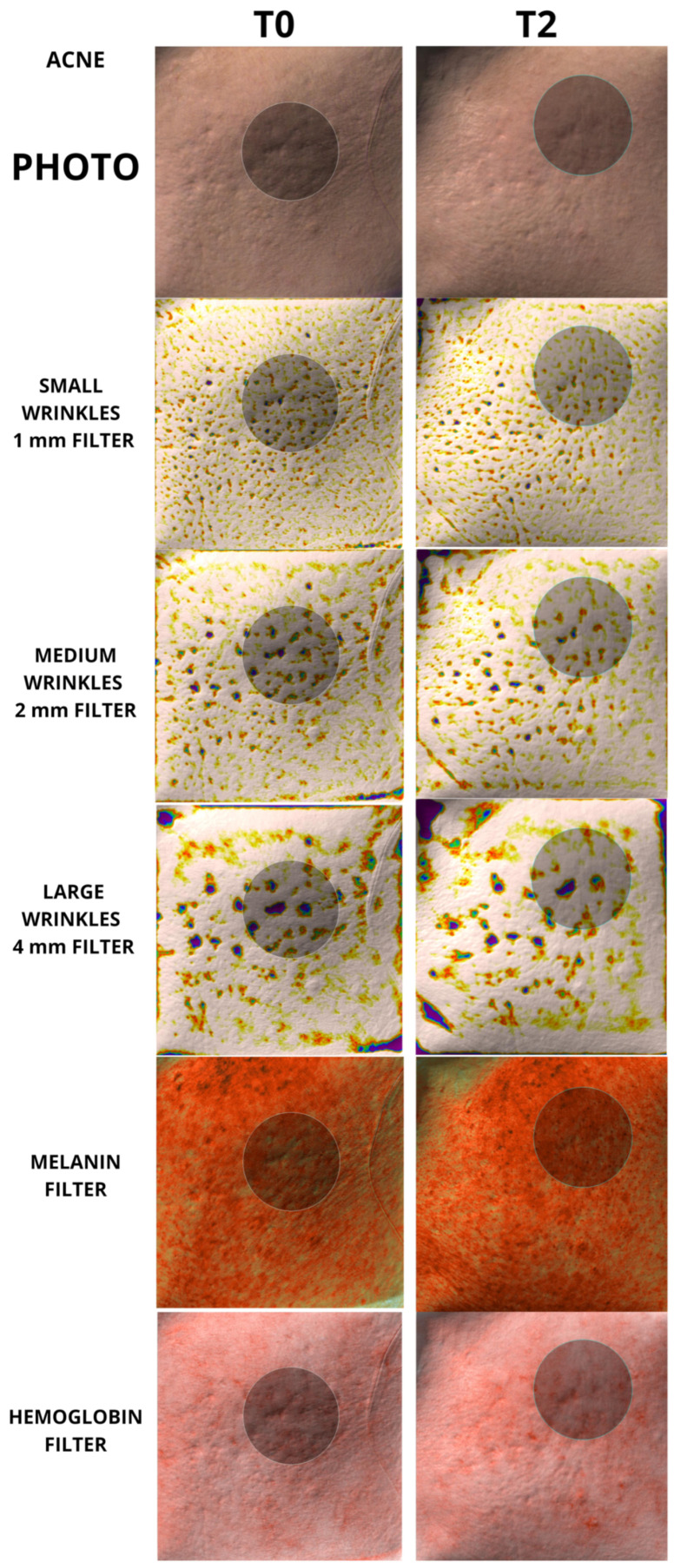
Picture taken from cheek area in a patient with acne scars. A 2.2 cm circle has been used for each measurement. Baseline and T2 pictures are shown.

**Table 1 biomedicines-12-00916-t001:** Descriptive statistics of the characteristics of the groups under analysis for the variables age and sex, expressed as mean and standard deviation for the continuous variables and frequency and percentage for the discrete variables.

Descriptive Statistics							
	Number of Patients	Female	Male	Age Minimum	Age Maximum	Mean	Std. Deviation
Normal group	50	42	8	30	70	49.76	10.916
Smoker group	20	18	2	31	70	52.5	13.233
Acne group	30	27	3	27	48	40.17	4.92

**Table 2 biomedicines-12-00916-t002:** SGAIS—Subject’s Global Aesthetic Improvement Scale.

SGAIS SCALE		
Rating		Description
**5**	very much improved	optimal cosmetic result
**4**	much improved	marked improvement in appearance from the initial condition, but not optimal for the patient. A touch up would slightly improve the result
**3**	improved	obvious improvement in appearance from the initial condition, but a touch up or retreatment is indicated
**2**	no change	the appearance is essentially the same as the original condition
**1**	worse	the appearance is worse than the original condition

**Table 3 biomedicines-12-00916-t003:** Summary table of data related to skin wrinkle depth measurements in the three groups. Fine wrinkles, medium wrinkles, and big wrinkles were analyzed separately for each area (cheek; eye; neck). The size of each group, chi-squares, and Friedman significance are shown for each group of measurements.

Friedman Test Wrinkle Depth						
	Normal Group		Smoker Group		Acne Group	
	χ Square	Sig. (2-Tailed)	χ Square	Sig. (2-Tailed)	χ Square	Sig. (2-Tailed)
CHEEK wrinkle depth (1 mm filter)	20.28	0.000	30.9	0.000	43.46	0.000
CHEEK wrinkle depth (2 mm filter)	36.48	0.000	40	0.000	56.26	0.000
CHEEK wrinkle depth (4 mm filter)	38.92	0.000	36.1	0.000	39.26	0.000
EYE wrinkle depth (1 mm filter)	43.72	0.000	31.6	0.000		
EYE wrinkle depth (2 mm filter)	20.52	0.000	30.1	0.000		
EYE wrinkle depth (4 mm filter)	17.76	0.000	12.7	0.002		
NECK wrinkle depth (1 mm filter)	6.76	0.034	40	0.000		
NECK wrinkle depth (2 mm filter)	0.52	0.771	18.1	0.000		
NECK wrinkle depth (4 mm filter)	9.64	0.000	40	0.000		

**Table 4 biomedicines-12-00916-t004:** Summary table of data related to melanin and hemoglobin measurements in the three groups for each area analyzed (cheek—eye—neck). The size of each group, chi-squares, and Friedman significance are shown for each group of measurements.

Friedman Test Hemoglobin Values (Antera 3D Scanner)						
	Normal Group		Smoker Group		Acne Group	
	χ Square	Sig. (2-Tailed)	χ Square	Sig. (2-Tailed)	χ Square	Sig. (2-Tailed)
**CHEEK wrinkle depth ** **(1 mm filter)**	0.579	0.749	34.9	0.000	46.33	0.000
**EYE wrinkle depth** **(1 mm filter)**	4.318	0.115	25.139	0.000		
**NECK wrinkle depth** **(1 mm filter)**	4.404	0.111	21	0.000		
**Friedman Test Hemoglobin Values (Antera 3D Scanner)**						
	**Normal Group**		**Smoker Group**		**Acne Group**	
	**χ Square**	**Sig. ** **(2-tailed)**	**χ Square**	**Sig.** **(2-tailed)**	**χ Square**	**Sig.** **(2-tailed)**
**CHEEK wrinkle depth (1 mm filter)**	24.121	0.000	37.696	0.000	10.41	0.005
**EYE wrinkle depth (1 mm filter)**	29.249	0.000	34.9	0.000		
**NECK wrinkle depth (1 mm filter)**	3.146	0.207	8.532	0.014		

**Table 5 biomedicines-12-00916-t005:** Kendall’s tau correlations between wrinkle depth for each filter (1 to 4 mm) and age for each group (normal—smokers—acne) (T0 = pre-injection measurements; Kendall’s tau correlations between melanin quantity and age for each group (normal—smokers—acne); Kendall’s tau correlations between hemoglobin quantity and age. T1 = first control measurements after 30 days; T2 = final control measurement after 60 days. Two-tailed significance data are shown; no correlations were found between variables and patient’s age.

Correlations between WRINKLE DEPTH Versus AGE										
	NORMAL GROUP				SMOKER GROUP			ACNE GROUP		
		T0	T1	T2	T0	T1	T2	T0	T1	T2
**CHEEK wrinkle depth (1 mm filter) versus patient’s age**	Sig. (2-tailed)	0.43	0.343	0.763	0.142	0.453	0.282	0.223	0.223	0.259
**CHEEK wrinkle depth (2 mm filter) versus patient’s age**	Sig. (2-tailed)	0.314	0.744	0.352	0.535	0.378	0.282	0.267	0.223	0.816
**CHEEK wrinkle depth (4 mm filter) versus patient’s age**	Sig. (2-tailed)	0.156	0.551	0.821	0.672	0.819	0.282	0.223	0.223	0.259
**EYE wrinkle depth (1 mm filter) versus patient’s age**	Sig. (2-tailed)	0.731	0.591	0.92	0.073	0.453	0.282			
**EYE wrinkle depth (2 mm filter) versus patient’s age**	Sig. (2-tailed)	0.86	0.172	0.352	0.181	0.063	0.282			
**EYE wrinkle depth (4 mm filter) versus patient’s age**	Sig. (2-tailed)	0.953	0.388	0.821	0.064	0.378	0.282			
**NECK wrinkle depth (1 mm filter) versus patient’s age**	Sig. (2-tailed)	0.687	0.663	0.586	0.063	0.063	0.282			
**NECK wrinkle depth (2 mm filter) versus patient’s age**	Sig. (2-tailed)	0.123	0.7	0.953	0.579	0.819	0.282			
**NECK wrinkle depth (4 mm filter) versus patient’s age**	Sig. (2-tailed)	0.214	0.063	0.563	0.214	0.033	0.563			
**Correlations between MELANIN QUATITY Versus AGE**										
	**NORMAL GROUP**				**SMOKER GROUP**			**ACNE GROUP**		
		**T0**	**T1**	**T2**	**T0**	**T1**	**T2**	**T0**	**T1**	**T2**
**CHEEK melanin quantity versus patient’s age**	Sig. (2-tailed)	0.192	0.207	0.33	0.26	0.057	0.409	0.23	0.355	0.986
**EYE melanin quantity versus patient’s age**	Sig. (2-tailed)	0.906	0.078	0.833	0.647	0.646	0.298			
**NECK melanin quantity versus patient’s age**	Sig. (2-tailed)	0.226	0.699	0.524	0.647	0.109	0.295			
**Correlations between HEMOGLOBIN QUANTITY Versus AGE**										
	**NORMAL GROUP**				**SMOKER GROUP**			**ACNE GROUP**		
		**T0**	**T1**	**T2**	**T0**	**T1**	**T2**	**T0**	**T1**	**T2**
**CHEEK melanin quantity versus patient’s age**	Sig. (2-tailed)	0.597	0.109	0.627	0.101	0.252	0.28	0.654	0.37	0.691
**EYE melanin quantity versus patient’s age**	Sig. (2-tailed)	0.43	0.374	0.614	0.06	0.591	0.326			
**NECK melanin quantity versus patient’s age**	Sig. (2-tailed)	0.358	0.393	0.31	0.073	0.102	0.31			

**Table 6 biomedicines-12-00916-t006:** Summary table showing the results of the analysis of the Face-Q Questionnaire. AGING APPRAISAL QUESTIONS = results of questionnaire “Aging Appraisal” concerning the subjective evaluation of the overall aging perceived by the patient; FACIAL APPEARANCE QUESTIONS = results of questionnaire “satisfaction with facial appearance” concerning the subjective evaluation of the overall face aspect perceived by the patient; APPRAISAL LINES OVERALL QUESTIONS = results of questionnaire “Appraisal of lines overall” concerning the subjective evaluation of the overall face wrinkles perceived by the patient; APPRAISAL NECK QUESTIONS = results of questionnaire “appraisal of the neck” concerning the subjective evaluation of the neck appearance perceived by the patient; SKIN QUALITY QUESTIONS = results of questionnaire “skin quality” concerning the subjective evaluation of the skin quality appearance perceived by the patient.

Face Q Questionnaire Results			
		Normal Group	
	Number of Patients	Chi-Square	Asymp. Sig
Aging appraisal questions	50	40.968	0.000
Facial appearance questions	50	70.235	0.000
Appraisal lines overall questions	50	47.582	0.000
Appraisal neck questions	50	48.042	0.000
Skin quality questions	50	49.834	0.000
		**Smokers Group**	
Aging appraisal questions	20	13.13	0.001
Facial appearance questions	20	27.103	0.000
Appraisal lines overall questions	20	28.43	0.000
Appraisal neck questions	20	15.233	0.000
Skin quality questions	20	13.899	0.001
		**Acne Group**	
Aging appraisal questions	30	24.122	0.000
Facial appearance questions	30	36.735	0.000
Appraisal lines overall questions	30	27.291	0.000
Appraisal neck questions	30	36.439	0.000
Skin quality questions	30	32.739	0.000

**Table 7 biomedicines-12-00916-t007:** SGAIS values expressed both as frequency and percentage. Pairwise comparison between the measurements performed during the first control (T1) and those performed during the second control (T2), with significance values for each group.

Wilcoxon Signed-Rank Test SGAIS QUESTIONNAIRE						
	NORMAL GROUP		SMOKER GROUP		ACNE GROUP	
Wilcoxon Signed-Rank Test	Z	Asymp. Sig. (2-Tailed)	Z	Asymp. Sig. (2-Tailed)	Z	Asymp. Sig. (2-Tailed)
	−3.186	0.001	−2.111 b	0.035	−4.244 b	0.000
	**NORMAL GROUP**		**SMOKER GROUP**		**ACNE GROUP**	
	**T1**	**T2**	**T1**	**T2**	**T1**	**T2**
**Very Much Improved**	16%	18%	15%	25%	0%	10%
**Much Improved**	12%	25%	10%	20%	0%	60%
**Improved**	70%	32%	70%	55%	76.6%	30%
**No Change**	2%	0%	5%	0%	23.3%	0%

## Data Availability

The datasets generated and analyzed during the current study are available from the corresponding authors upon reasonable request.

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
