# Peer review of "New Regenerative and Anti-Aging Medicine Approach Based on Single-Stranded Alpha-1 Collagen for Neo-Collagenesis Induction: Clinical and Instrumental Experience of a New Injective Polycomponent Formulation for Dermal Regeneration"

_biomedicines, 2024, doi:10.3390/biomedicines12040916_

Round 1
Reviewer 1 Report
Comments and Suggestions for Authors
Abstract
Please add a "take-home" message in the abstract
Introduction
Please clearly explain the novelty of your work.
Describe the novelty of the article made by the author. If there is something new in this manuscript, please highlight it more clearly in the introduction section.
The limitations of similar prior studies must be explained in the introduction section to highlight the research gaps that the current study aims to fill.
Please explain what the authors did to prove the hypothesis of your study at the end of the introduction section.
There was no proper cite the related references in the introduction section such as Line 51-55, 59-61, and 74-90.
The authors mentioned “a new medical device” but did not mention the components of this new formulation.
The authors need to elaborate on previous literature, including their previous work, recruiting similar conceptualization and/or methodology to address this research question. Thus, please review the literature on similar published works.
Materials and Methods
Please describe what the authors wanted to measure before each experiment.
No statistical analysis section.
Statistical analysis is needed to add more specific information.
Results
The quality of Figure 1-3 needs to be improved.
The results were mainly analyzed using statistical analysis tools. Thus, it would be clearer to emphasize the analysis tools and assumptions used in each experiment.
Discussions
As outcomes must be compared to similar past research, the authors already added the previous research. However, the discussion in the present article needs to improve to become more comprehensive.
Conclusions
Can the authors draw a conclusion/explanation (supported by evidence)
General comments
Please revisit the entire manuscript for grammar and typo issues.
Improve the quality of Figure
Author Response
THANK YOU FOR YOUR VALUABLE SUGGESTIONS.
BELOW ARE THE CHANGES REQUIRED:
ABSTRACT
- The take home message was included in the abstract as recommended.
Introduction
- the innovations brought by our work, i.e. being the first in vivo clinical study on patients of the capabilities of this new type of recombinant collagen, have been highlighted as requested in the introduction.
- it was highlighted that there are no other previous works for this type of medical device and that this work is complementary to the discoveries of the work done previously on normal and aged human fibroblasts.
- we explained better in the text how we verified our hypotheses using objective methods such as the Antera 3D scanner and subjective methods such as validated questionnaires.
- we have corrected the references to the lines indicated by the reviewers.
- we have included in the text the precise formulation of the medical device indicating the concentrations of the molecules contained in it.
- we reworked the literature by adding the references of previous studies that used the resources and technologies we used for our work.
Materials and methods
- explanations regarding the measures that were evaluated before each experiment have been included in the text.
- the statistical analysis had been included in the text, but now, as per the reviewer's request, a separate paragraph has been created with the explanation of the statistical analysis and the tests carried out.
Results
- We modified images 1-2-3 by removing less significant data and making the results of the box plots clearer and more visible
- we have emphasized and explained better in the text the statistical tools used and the rationale for their use.
Discussion
- We have modified the discussion of the study by broadening it and making our objectives clearer.
Conclusion
- Conclusions were expanded according to the reviewers' instructions.
- We have also corrected all grammatical and typographical errors.
- we have also edited and improved the quality of all images
Reviewer 2 Report
Comments and Suggestions for Authors
This is a very interesting approach to regenerative and anti-aging medicine. The contents are well organized and very attractive. However, in order to enhance the value of this article, please consider the following review comments.
#1. The words ‘New Medical Device’ and ‘Instrumental’ are not appropriate for this study of the proposed injectable collagen material with hyaluronic acid and carboxymethyl cellulose. Because those words will give an image of a hardware device. Instead of those, ‘New Injective Medicine’ or ‘New Injective Material’ are recommended.
Consequently, the title can be changed to ‘New Regenerative and Anti-Aging Medicine Approach Based on Single Strand Alpha-1 Collagen for Neo-Collagenesis Induction: Clinical Experience and Statistical Analysis of a New Injective Medicine for Dermal Regeneration’.
#2. Careful proofreading is required in the main text in terms of misspelling, coexistence of British/American spelling, font error.
Line 72 fibres (British spelling), fibers (American spelling)
Line 108 49.76 ±10.916 (Font Error for Mean ± Std. Deviation), some others are the same.
Line 167 ‘repeated again’ erase ‘again’ (double expression)
Line 305 P<0.000 (standard format)
Line 306 P=0.771 (standard format)
Line 309 analized analyzed
Line 346 moderare moderate
Line 349 intravasal intravenous?
Line 455 ialuronic hyaluronic
Author Response
THANK YOU FOR YOUR VALUABLE SUGGESTIONS.
BELOW ARE THE CHANGES REQUIRED:
• the word new “medical device” has been replaced with the term deemed more appropriate by the reviewer. The word “instrumental” has been removed from the text, similarly, the title of the study has also been changed. However, it must be underlined that the choice of the term "medical device" was not chosen by the authors, but it is simply in this category that injectable products in Italy fall. This is the reason why this term was used. Injective Polycomponent formulation was chosen by the authors for the title change because it is the same used by Augello et al who studied the compound in vitro.
• A rigorous search for typos and grammatical errors was made throughout the text.
• Line 72 has been fixed
• Line 108 has been corrected
• Line 167 has been corrected
• Line 305 has been corrected
• Line 306 has been corrected
• Line 309 has been corrected
• Line 346 has been corrected
• Line 349 has been corrected
• Line 455 has been corrected
Reviewer 3 Report
Comments and Suggestions for Authors
This research paper «New Regenerative and Anti-Aging Medicine Approach Based on Single Strand Alpha-1 Collagen for Neo-Collagenesis Induction: Clinical and Instrumental Experience of a New Medical Device for Dermal Regeneration» is devoted to the study of the effectiveness of a new medical device in regenerative medicine for skin regeneration and rejuvenation. The study involved 100 people with various skin diseases. After testing in several stages, it can be concluded that this medical device effectively stimulates skin regeneration and improves its quality and structure. However, some comments were highlighted:
1. The literary references on lines 68, 180, 211, 392-393 and 444 are incorrectly designed. If there are several links, then it is worth writing them separated by commas or dashes in one bracket, rather than dividing each link into a separate bracket. For example: [1, 2] or [1-5].
2. Lines 191-207. On these lines there is a description of all the questionnaires used, in which the general breakdown and summation of points is probably worth putting the general information at the very beginning, so that the same information is not repeated in each paragraph.
3. Line 274-275. In the same sentence, there is a reference in "Table 1" twice.
4. Line 339. The abbreviation "k" should be capitalized, not lowercase.
5. Check the links to the text by text, since in many cases there is no space between the word and the bracket, for example, the line 48 "... %[1]".
6. The "References" item (page 471). Remove the first and second paragraphs, as this is a description of how to properly design the list of references, which is presented below.
Author Response
REVIEWER 3:
THANK YOU FOR YOUR VALUABLE SUGGESTIONS.
BELOW ARE THE CHANGES REQUIRED:
• The bibliographical references have been corrected following your advice.
• the description of the questionnaires has been rewritten to be more understandable and to eliminate redundancies
• The error relating to the reference to Table 1 has been corrected.
• The abbreviation has been corrected.
• space between words and brackets have been added throughout the text.
• LINES FROM 471 TO 479 HAVE BEEN DELETED (UNFORTUNATELY IT WAS A PRINTING MISTAKE)
Reviewer 4 Report
Comments and Suggestions for Authors
In the manuscript (ID: biomedicines-2900672), the authors present manuscript concerning the study, that explores the efficacy of a novel medical device in regenerative medicine, for skin regeneration and rejuvenation. The manuscript is interesting, however it should be improved.
The manuscript requires thorough editorial improvement. I have a major concerns that can be easily addressed:
1) The abbreviation should be explained for the first time in the text and not repeated explanation once again. Please check the text.
2) The text should be carefully checked and corrected (small letters in the beginning of sentence, lack of dots, dots in wrong place in sentence etc)
3) Please correct: TGF-b1 or TGF-β1.
4) The Table 5 should be presented in more clear way. The other tables also should be improved.
5) The text from line 472 to 479 should be removed.
Author Response
REVIEWER 4:
THANK YOU FOR YOUR VALUABLE SUGGESTIONS.
BELOW ARE THE CHANGES REQUIRED:
• the take home message has been added to the abstract
• Abbreviations have been corrected throughout the text, indicating at the beginning of the manuscript what the abbreviations used in the text correspond to.
• The text has undergone an extensive grammatical and punctuation review process, typing errors (such as misplaced periods) have been eliminated.
• TGF-β1 has been corrected throughout the manuscript.
• Table 5 has been completely redesigned maintaining the original values, in order to make it more understandable. The other tables have also been rethought and redesigned to be clearer. The table legends have also been rewritten.
• LINES FROM 472 TO 479 HAVE BEEN DELETED (UNFORTUNATELY IT WAS A PRINTING MISTAKE)
• All figures have been revised and the quality has been improved.
Reviewer 5 Report
Comments and Suggestions for Authors
ABSTRACT
- The authors should provide more information about the tested product in the abstract section. Please note that the name of the product and its formulation are not presented in this section.
INTRODUCTION
- The authors should include the appropriated references for the texts presented at the introduction. Some paragraphs are missing the appropriated references.
- Please improve the presentation of the introduction. Some paragraphs are too short, with only two lines.
MATERIAL AND METHODS
- Please consider to provide the ‘inclusion and exclusion criteria’ and the ethic statement before the presentation of the patient selection.
- The statistical analysis section is missing. This should not be provided at begging of the results section.
Author Response
REVIEWER 4:
THANK YOU FOR YOUR VALUABLE SUGGESTIONS.
BELOW ARE THE REQUESTED CHANGES:
• The abstract has been updated according to the reviewer's recommendations.
• The appropriate references needed in the introduction have been added.
• The introduction has been expanded and made clearer. Furthermore, as requested, all paragraphs have been revised, and the smaller ones have been modified or merged.
• Inclusion and exclusion criteria and ethical statements have been moved to the beginning of the paragraph on the presentation of patient selection.
• The paragraph on statistical analysis has been written and included in the materials and methods, explaining which tests were used.
Round 2
Reviewer 1 Report
Comments and Suggestions for Authors
No more comments
Author Response
Thanks for your previous reviews. they were very helpful in improving our manuscript
Reviewer 4 Report
Comments and Suggestions for Authors
The manuscript was corrected but not in all aspects. The tables should be corrected, e.g. smaller font and line spacing, so that they fit legibly on the page. In an abstract, the first sentence begins with a lowercase letter. The text should be justified. Abbreviations should be explained when they are first used in the text.
Author Response
Thanks for your previous reviews. they were very helpful in improving our manuscript. we have corrected the tables, changing the spacing and using a smaller font. we corrected the lowercase in the abstract, we justified all the text. we have also corrected the abbreviation in the abstract.
Reviewer 5 Report
Comments and Suggestions for Authors
The authors have changed the manuscript following the reviewer suggestions.
Author Response
Thanks for your previous reviews. they were very helpful in improving our manuscript.